# Study on Soil Moisture Characteristics in Southern China Karst Plant Community Structure Types

**Xing Gu** [1,2]**, Kangning Xiong** [1,2,*]**, Chenxu Wu** [1,2] **and Ding Luo** [1,2,3]

1   School of Karst Science, Guizhou Normal University, Guiyang 550001, China
2   State Engineering Technology Institute for Karst Desertification Control, Guiyang 550001, China
3   School of Tourism and Resources Environment, Qiannan Normal University for Nationalities, Duyun 558000, China
*   Correspondence: xiongkn@gznu.edu.cn

**Abstract:** An understanding of soil moisture variation patterns under different plant community structures is crucial for the restoration of vegetation in southern China karst regions. Therefore, four plant community structure types: arbor + herb (AH), shrub + herb (SH), arbor + shrub + herb (ASH), and herb (H), were selected as the research objects. A soil moisture sensor was used to monitor the soil moisture content in the 0–70 cm soil layer, to analyze the variation characteristics of soil moisture content and to explore the differences under different plant community structure types. The results indicate that: (1) A total of 31 plant species in 31 genera and 18 families were recorded, among which herbs were the most abundant. There were significant differences in diversity indexes for ASH and H. The differences between Shannon–Wiener index and Simpson index of AH and H were significant, and between AH and SH in Pielou index and Simpson index were statistically significant. The Pielou index between SH and H was significantly different. (2) There were significant differences in soil water content among the four plant community types, showing SH > AH > H > ASH. The variation of soil moisture was consistent with the trend of rainfall, with the surface soil moisture more sensitive to rainfall events, and the deeper soil moisture had a lag. The Shannon–Wiener index was significantly and positively correlated with the Simpson index and the Margalef index. There was a highly significant positive correlation between Margalef index and Simpson index. The 20–30 cm soil layer was highly negatively correlated with the Margalef index and significantly and negatively correlated with the Shannon–Wiener index and the Simpson index. (3) The response of soil moisture of plant community structure types to light rain event was not obvious. The growth rates of soil water content in the 30–50 and 50–70 cm layers of the SH were higher in moderate rain event than that in heavy rain event, and H, AH, and ASH had larger growth rates in heavy rain events. The results provide a reference for the effective utilization of soil and water resources and the restoration of vegetation, as well as for plant community structure configuration decisions in the southern China karst region.

**Keywords:** soil moisture; plant community structure; rainfall; Karst; the restoration of vegetation





## 1. Introduction

Soil moisture is a key factor in controlling soil–plant energy balance and nutrient cycling. It plays an important role in determining ecosystem composition and function and indirectly or directly participates in the processes of runoff, soil evaporation and plant transpiration [1–3]. The karst of southern China, centered on the Guizhou Plateau, is a fragile karst zone with the widest distribution, the largest types, the most complex ecological environment and the largest exposed area of carbonate rocks among the three karst regions in the world [4–6]. Rainfall in the area is relatively abundant. However, soil moisture is still a major factor limiting ecological restoration and sustainable land use in the area due to the extensive distribution of bedrock, strong chemical dissolution, and

the special geological structure that forms a soil erosion pattern with surface loss and underground leakage. Rainfall is an important factor in soil moisture variation. The process and efficiency of rainfall conversion into soil moisture are affected by the type of plant community structure overlying the soil, topography and soil properties, and the special binary three-dimensional spatial structure in the karst area leads to the complicated soil moisture infiltration mechanism [7–9]. Soil moisture plays an important role in vegetation growth, and vegetation can influence soil moisture and its response to rainfall through many complex and interactional hydrological processes. Plant communities can affect soil moisture change by altering canopy interception, plant root absorption, and soil physical and chemical properties. However, canopy and root distribution characteristics of different plant community structure types are different. When the plant community structure is in good condition, it can effectively reduce the surface runoff and soil nutrient loss and increase the absorption capacity of soil moisture [10–13].

Plant community structure is the spatial and temporal configuration of the various plants in the community, which can be expressed as the interaction between plants and the environment. It has a certain spatial structure and affects a series of characteristics of the community [14,15]. Species diversity can reflect community structure type, organization level, development stage, stability degree, and habitat difference [16]. At present, studies on plant community structure types and soil moisture mainly focus on the differences of vertical distribution of soil moisture in different community structure types, the response characteristics to rainfall and the influence of species diversity on soil moisture [17–20]. Wang et al. monitored the soil moisture content of different plant community structure types in the Loess Plateau and found that soil moisture was mainly recharged by rainfall events, and community structure types affected soil moisture infiltration [21]. Cheng et al. monitored the soil moisture characteristics of three plant community structure types: natural shrub, *Robinia pseudoacacia* plantation and secondary forest, and found that natural shrub had high adaptability to local precipitation [22]. Qin et al. explored the response of plant community structure characteristics to soil moisture content in the Badain Jaran Desert and showed that soil moisture content was highly significantly negatively correlated with the Simpson dominance index and highly significantly positively correlated with the Shannon–Winner diversity index, Simpson diversity index and Alatato evenness index [23]. Zhang et al. investigated and analyzed the community structure characteristics of the semi-arid grassland community on the Loess Plateau, to study the influence of soil moisture content changes on community structure, and showed that soil moisture content was positively correlated with species diversity and richness [24]. At present, studies on the relationship between plant community structure type and soil moisture have mainly been carried out in non-karst areas, while soil moisture studies in karst areas are mostly focused on the profile characteristics and spatial patterns of soil moisture under land use types [25–27]. For example, Li et al. studied the soil moisture profile characteristics of farmland and grassland in karst depression [26]. Zhang et al. studied the spatial pattern of field surface soil moisture in a karst depression area of Huanjiang County, Guangxi Province, of southwest China [27]. Chen et al. studied the spatial and temporal dynamics of shallow soil moisture on karst slopes under different land uses in northwestern Guangxi [25]. Therefore, the different plant community structure types and soil moisture characteristics variation in karst areas still pose a number of worthy scientific questions. The assessment of soil moisture dynamics and precipitation response under plant community structure types in the southern China karst can provide decision support for ecosystem restoration, plant community structure configuration patterns and land resource optimal utilization in karst areas.

In this study, we aimed to describe the soil moisture variation patterns of different plant community structure types in karst areas. The objectives were to explore (1) the characteristics of plant diversity and its relationship with soil moisture in different plant community structure types, (2) the differences in the spatial and temporal distribution of

soil moisture under different plant community structure types, and (3) the response of soil moisture to different rainfall events of different plant community structure types.

## 2. Materials and Methods

### 2.1. Study Area

This study was conducted at the Bijie Salaxi, Guizhou, China (105°01′10″–105°08′39″ E, 27°11′08″–27°17′30″ N), and belongs to the Liuchong River basin in Qixingguan District, with a total area of 86.27 km² and an annual average temperature of 12 °C (Figure 1). It has a humid subtropical monsoon climate, with an altitude of 1509~2180 m and abundant rainfall. The annual average rainfall is 984 mm, but the rainfall distribution is uneven and mainly concentrated in the months of May to September. The karst area accounts for 74.25% of the total area of the study area, and potential, mild, moderate and severe karst desertification are all distributed in the study area. The topography of the study area is fragmented, and the soils are predominantly yellow soil with a small proportion of yellow-brown soil. The native vegetation in the area mainly includes Masson pine forest, rhododendron forest, grass-land, etc., but human disturbance has basically destroyed the native vegetation. The primary vegetation in the area mainly consists of *Pinus massoniana*, *Rhododendron* and grasslands, but human disturbance has basically destroyed the primary vegetation, and the plants are mainly secondary vegetation, the economic forests are mainly planted with *Juglans regia* and *Ribes burejense*, and the crops are mainly *Zeamays* and *Solanum tuberosum* [28].

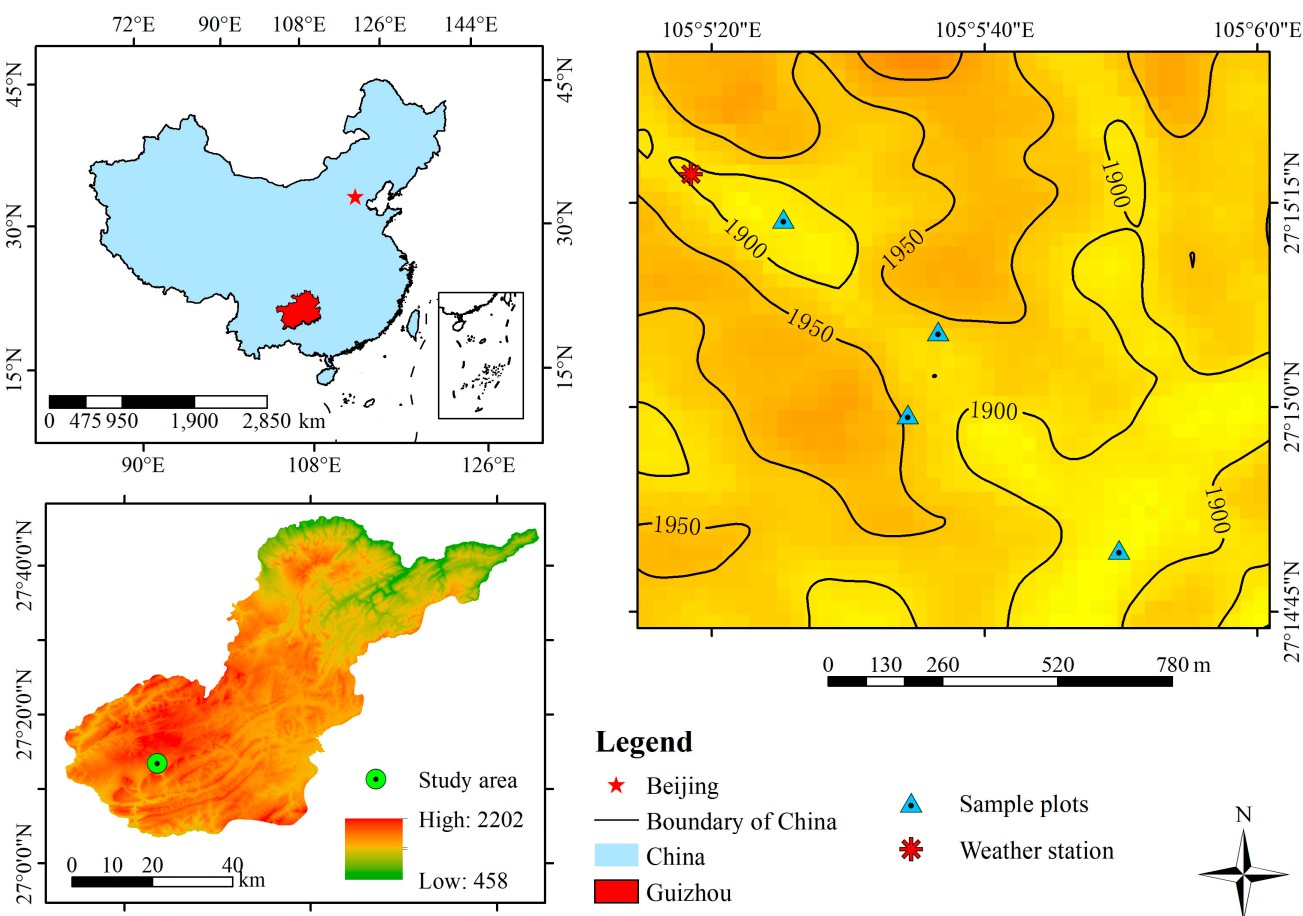

**Figure 1.** Geographical location and sample plots distribution of study area.

### 2.2. Soil Moisture Monitoring

Four community structure types of herb (H), arbor + herb (AH), shrub + herb (SH), and arbor + shrub + herb (ASH) with basically the same slope position, slope direction, slope and altitude were selected as test plots in the study area. In each sample plot, soil moisture sensors ( ECH$_2$O-5TE, METER Group Inc., Pullman, WA, USA, accuracy 0.03 m$^3$/m$^3$) were installed to monitor the soil moisture content in 0–10, 10–20, 20–30, 30–50 and 50–70 cm soil layers. Data were collected using a data collector (EM50, METER Group Inc., Pullman, WA, USA) at a frequency of 30 min/time. In addition, the monitoring data used in this paper are for 4 months from July to October 2021, with a small meteorological observation (ATMOS, METER Group Inc., Pullman, WA, USA) in the study area to monitor rainfall, temperature and other indicators at a frequency of 30 min/time.

### 2.3. Research Method

#### 2.3.1. Experimental Design

The sample method was used to conduct field investigations, and one sample survey of the four plant community structures was conducted during July and August 2021. In each sample plot, a 10 × 10 m square was set up and replicated in three groups. All woody plants ≥1.5 m in height were classified as arbor, and the species, height, number, DBH and crown width of trees in the sample were investigated and recorded. Three 5 × 5 m quadrates were set in the larger quadrates along the diagonal. All woody plants with tree height <1.5 m were investigated in the 5 × 5 m quadrates, and their species, number, height, crown width and coverage were recorded. In the four corners of the 10 × 10 m sample squares, 1 × 1 m sample squares were set up for herbaceous survey. The survey contents included the species, abundance, height and coverage of herbaceous plants, etc. The basic condition of the sample plots is shown in Table 1. During the investigation, the information of the sample plots was recorded. GPS was used to obtain location information and record various environmental factors such as altitude, slope and slope direction in the sample plots.

**Table 1.** Basic information of the sample plots.

| Sample Site | Vegetation Coverage/% | Altitude/m | Soil Bulk Density (g/cm$^3$) | Structure Type | Dominant Species | Porosity (g/cm$^3$) |
|---|---|---|---|---|---|---|
| H | 85 | 1870 | 1.44 | Herb | *Lolium perenne, Trifolium repens* | 46.43 |
| AH | 83 | 1865 | 1.42 | Arbor + Herb | *Juglans regia, Artemisia argyi* | 47.18 |
| SH | 70 | 1872 | 1.41 | Shrub + Herb | *Ribes burejense, Lolium perenne* | 47.42 |
| ASH | 78 | 1890 | 1.30 | Arbor + Shrub + herb | *Hypericum monogynum, Pyracantha fortuneana, Lolium perenne* | 51.19 |

#### 2.3.2. Calculation Formula and Data Processing

Based on plant community survey, meteorological data and continuous dynamic monitoring of soil moisture, the characteristics of plant community species diversity, vertical distribution characteristics of soil moisture in different plant community structure types and the response of soil moisture to rainfall were analyzed. In this study, the Shannon–Wiener index (H′), Margalef richness index (R), Pielou evenness index (E) and Simpson diversity index (D) of α-diversity were used to reflect the species composition and richness of the community [29–31] (Table 2). SPSS 23.0 was used for data analysis, and Origin 2019 was used for graphing.

**Table 2.** The calculation formula of $\alpha$-diversity index.

| Index | Formula |
|---|---|
| Shannon–Wiener Diversity Index | $H\prime = -\sum\limits_{i=1}^{n} p_i ln p_i$ |
| Margalef Richness Index | $R = (S-1)/lnN$ |
| Pielou Evenness Index | $E = H'/lnS$ |
| Simpson Diversity Index | $D = 1 - \sum\limits_{i=1}^{n} (p_i)^2$ |

## 3. Analysis of Results

### 3.1. Plant Community Structure Composition

#### 3.1.1. Vegetation Composition of the Sample Plots

The results showed that there was a total of 31 species in 31 genera and 18 families in this study, including 6 species in 6 genera of Compositae, 4 species in 4 genera of Gramineae, 4 species in 4 genera of Rosaceae, 2 species in 2 families of Urticaceae, 2 species in 2 families of Rubiaceae, and the rest are all 1 family and 1 species (Table 3). In the sample plots with different plant community structure types, H had the highest number of species composition, followed by SH, AH and ASH. The plant community structure was relatively simple in each sample plot, with more single families, single genera and single species. The species composition of shrub layer and arbor layer is relatively monotonous, with plants mainly in Rosaceae, Juglandaceae and Salicaceae. The herbaceous layer was mostly perennial herbaceous, accounting for 66.7% of the survey species.

#### 3.1.2. Species Diversity Characteristics of Different Plant Community Structure Types

According to the survey results (Figure 2), it can be seen that the Margalef richness index (R), Shannon–Wiener diversity index (H') and Simpson diversity index (D) of different plant community structure types showed H > ASH > SH > AH. The Margalef richness index (R) and Shannon–Wiener diversity index (H') of ASH were significantly higher than that of AH ($p < 0.05$), and the Shannon–Wiener diversity index (H') and Simpson diversity index (D) of H were significantly higher than that of AH ($p < 0.05$). Although the richness index and diversity index of H were higher, the plant community structure was relatively simple, and the vertical stratification of the community was not obvious. The Simpson diversity index (D) for SH and ASH differed from AH in a statistically significant way. In terms of the Pielou evenness index (E), plant community structure types showed SH > ASH > H > AH. The Pielou evenness index (E) of SH and ASH was significantly higher than that of AH and H ($p < 0.05$).

### 3.2. Soil Moisture Characteristics of Plant Community Structure Types

#### 3.2.1. Vertical Distribution Characteristics of Soil Moisture

The soil moisture content of the four plant community structure types was significantly different in the five soil layers ($p < 0.05$). The SH has a significantly higher soil moisture content than H and ASH structure types in the 0–10 cm soil layers and a significantly higher soil water content than other plant community structure types in the 30–50 and 50–70 cm (Figure 3B,E,F). The soil water content of H was significantly higher than that of SH and ASH in the 10–20 cm soil layer (Figure 3C), and the soil moisture content of AH in the 20–30 cm soil layer was significantly higher than that of other plant community structure types (Figure 3D), which ultimately led to significant differences among the four plant community structure types in the 0–70 cm range, showing SH > AH > H > ASH (Figure 3A). The soil moisture of H tends to increase and then decrease from the surface to the deeper layers, with the most moisture content in the 20–30 cm soil layer, while the soil moisture content of AH and SH tends to decrease and then increase from the surface to the deeper

layers, with the most moisture content in the 50–70 cm soil layer. The soil moisture content of ASH soils was relatively low in all layers.

**Table 3.** The species composition of plant community.

| Families | Genera | Species | Community Structure Type | Life Form |
|---|---|---|---|---|
| Compositae | *Artemisia* | *Artemisia argyi* | H/SH/SH/ASH | perennial herbaceous |
| | *Cirsium* | *Cirsium japonicum Fisch* | H/SH | perennial herbaceous |
| | *Carpesium* | *Carpesium abrotanoides* | H/AH/ASH | perennial herbaceous |
| | *Arctium* | *Arctium lappa* | H/AH/SH | biennial herbs |
| | *Erigeron* | *Erigeron annuus* | H/SH/ASH | annual herbaceous |
| | *Conyza* | *Conyza canadensis* | AH/SH/ASH | annual herbaceous |
| Gramineae | *Secale* | *Lolium perenne* | H/SH/SH/ASH | perennial herbaceous |
| | *Imperata* | *Imperata cylindrica* | H/SH/ASH | perennial herbaceous |
| | *Setaria* | *Setaria viridis* | H/AH/SH/ASH | annual herbaceous |
| | *Pogonatherum* | *Pogonatherum crinitum* | H/SH/ASH | perennial herbaceous |
| Rosaceae | *Agrimonia* | *Agrimonia pilosa* | H/AH/SH/ASH | perennial herbaceous |
| | *Fragaria* | *Fragaria nilgerrensis Schlecht* | AH/ASH | perennial herbaceous |
| | *Rosa* | *Ribes burejense* | SH | shrub |
| | *Pyracanth* | *Pyracantha fortuneana* | SH/ASH | shrub |
| Urticaceae | *Urtica* | *Urtica fissa* | H/SH/ASH | perennial herbaceous |
| | *Boehmeria* | *Boehmeria spicata* | AH/ASH | perennial herbaceous |
| Rubiaceae | *Galium* | *Trifolium repens* | H/SH/SH/ASH | perennial herbaceous |
| | *Borreria* | *Borreria latifolia* | H/AH/ASH | perennial herbaceous |
| Pteridaceae | *Pteris* | *Pteris cretica* | H/SH | perennial herbaceous |
| Thelypteridaceae | *Parathelypteris* | *Parathelypteris glanduligera* | H | perennial herbaceous |
| Geraniaceae | *Geranium* | *Geranium wilfordii Maxim* | H/AH/ASH | perennial herbaceous |
| Caryophyllaceae | *Stellaria* | *Stellaria vestita* | H/SH/SH/ASH | perennial herbaceous |
| Leguminosae | *Vicia* | *Vicia sepium* | H/SH/SH/ASH | perennial herbaceous |
| Coriariaceae | *Coriaria* | *Coriaria nepalensis* | SH/ASH | shrub |
| Hamamelidaceae | *Corylopsis* | *Corylopsis sinensis* | ASH | shrub |
| Alangiaceae | *Alangium* | *Alangium chinense* | ASH | shrub |
| Juglandaceae | *Juglans* | *Juglans regia* | AH/ASH | arbor |
| Salicaceae | *Populus* | *Populus* sp. | AH | arbor |
| Anacardiaceae | *Toxicodendron* | *Toxicodendron vernicifluum* | AH/ASH | arbor |
| Clusiaceae | *Hypericum* | *Hypericum monogynum* | ASH | arbor |
| Pinaceae | *Pinus* | *Pinus massoniana* | ASH | arbor |

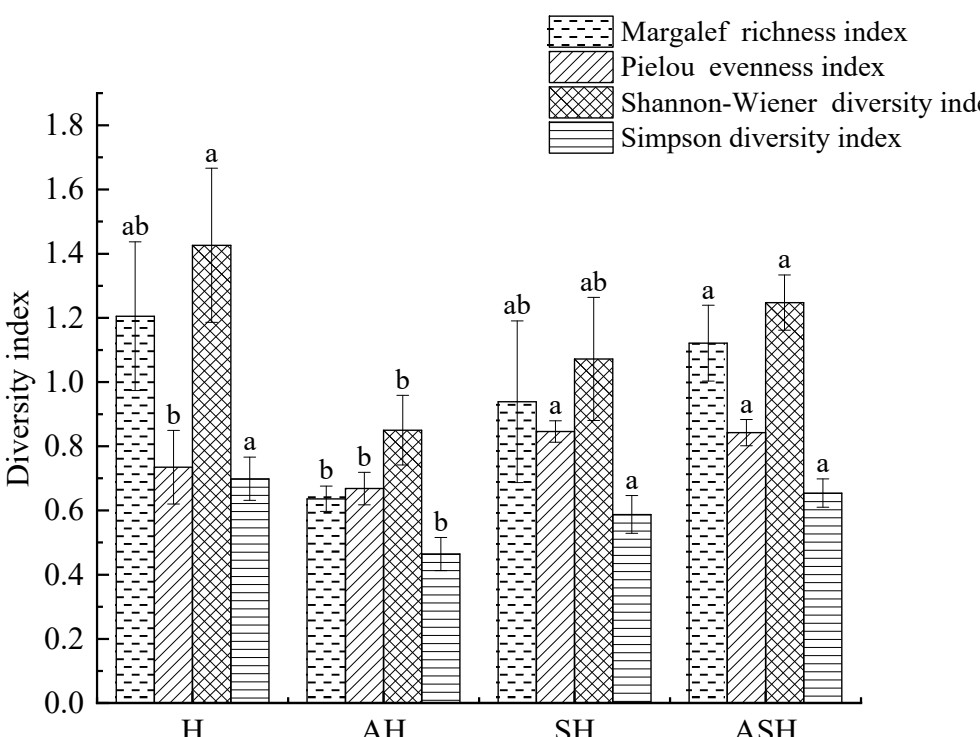

**Figure 2.** Species diversity index of plant community structure types. Different lowercase letters indicate significant differences of diversity index in different plant community structure types ($p < 0.05$).

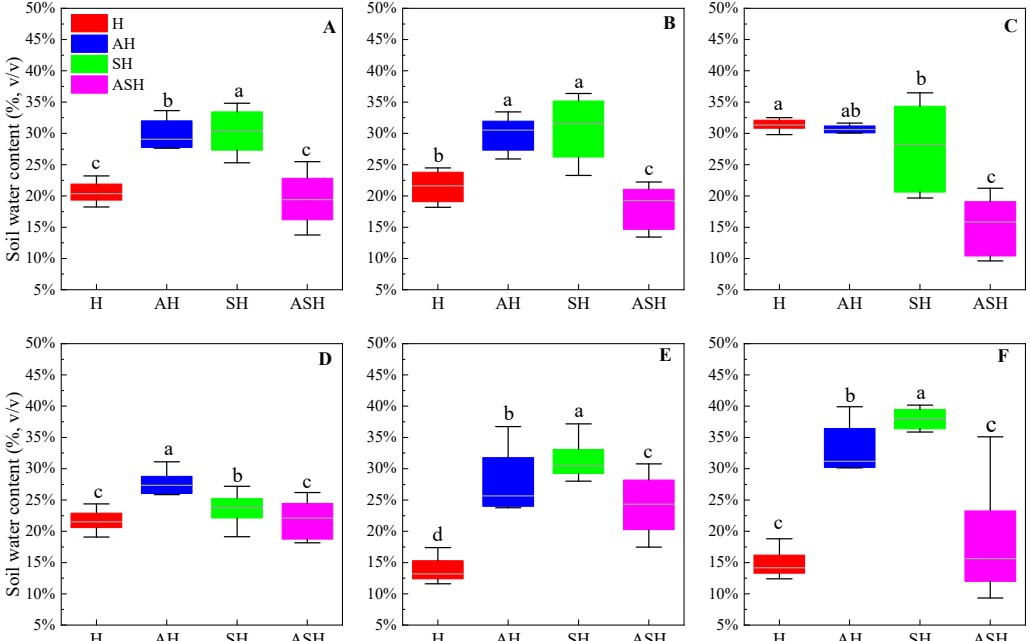

**Figure 3.** Comparison of soil water content for 0–70 cm. (**A**) 0–70 cm, (**B**) 0–10 cm, (**C**) 10–20 cm, (**D**) 20–30 cm, (**E**) 30–50 cm, (**F**) 50–70 cm. Different lowercase letters indicate significant differences of soil water content in the different plant community structure types of the same soil layer ($p < 0.05$).

3.2.2. Temporal Distribution Characteristics of Soil Moisture

The variation of soil moisture content in each soil layer of the four plant community structure types was consistent with that of rainfall, and the soil moisture increased in different degrees after rainfall events (Figure 4). The soil moisture of the surface was

more sensitive to rainfall, while there was a significant lag in deep soil moisture response (Figure 4A,E). The soil moisture content of SH and AH were at a high level in all soil layers, and the soil moisture content of H in 30–70 cm was lower than that of other plant community structure types (Figure 4D,E). The soil moisture content of four plant community structure types showed the same trend in the 0–10 and 20–30 cm layers. The soil moisture variation of H was relatively gentle compared with other plant community structure types in all soil layers.

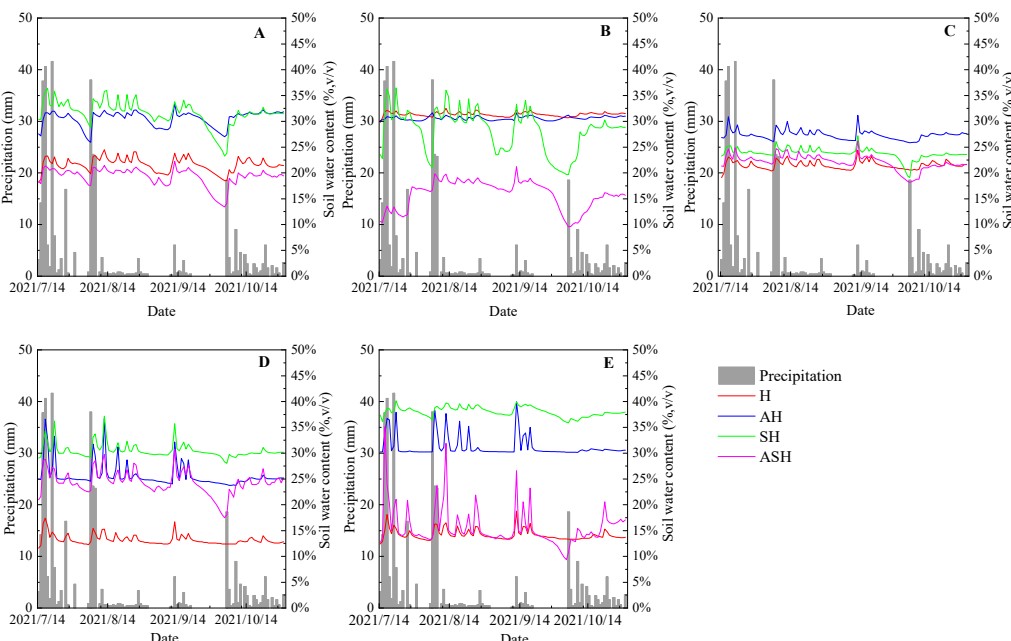

**Figure 4.** Daily variations of soil moisture content for 0–70 cm. (**A**) 0–10 cm, (**B**) 10–20 cm, (**C**) 20–30 cm, (**D**) 30–50 cm, (**E**) 50–70 cm.

### 3.2.3. Relationship between Species Diversity and Soil Moisture in Different Plant Community Structure Types

Through the analysis of the correlation between species diversity and soil moisture of each plant community structure type (Table 4), it can be seen that soil moisture and different plant community structure types have different degrees of differences. The Pielou evenness index was not significant to Margalef richness index, Shannon–Wiener diversity index and Simpson diversity index. The Shannon–Wiener diversity index has a significant positive correlation with Margalef richness index and Simpson diversity index ($p < 0.05$). There was a highly significant positive correlation between Margalef richness index and Simpson diversity index ($p < 0.01$). The soil moisture content in 0–10 and 10–20 cm soil layers was negatively correlated with the diversity index and did not reach a significant level. The Pielou evenness index was positively correlated with soil moisture in 30–50 and 50–70 cm soil layers, but negatively correlated with other diversity indexes. The 20–30 cm soil layer showed highly significant negative correlations ($p < 0.01$) with the Margalef richness index and Simpson diversity index and showed significant negative correlation with the Shannon–Wiener diversity index ($p < 0.05$), which was related to the type of plant community structure and root distribution. The Pielou evenness index was not significantly correlated with all soil layers.

**Table 4.** Correlation coefficients between species diversity index and soil moisture content at different depths.

| Index | | Margalef Index | Pielou Index | Shannon–Wiener Index | Simpson Index | Soil Moisture Content | | | |
|---|---|---|---|---|---|---|---|---|---|
| | | | | | | 0–10 cm | 10–20 cm | 20–30 cm | 30–50 cm |
| Pielou index | | 0.477 | | | | | | | |
| Shannon–Wiener index | | 0.981 * | 0.298 | | | | | | |
| Simpson index | | 0.998 ** | 0.468 | 0.983 * | | | | | |
| Soil moisture content | 0–10 cm | −0.791 | −0.182 | −0.799 | −0.757 | | | | |
| | 10–20 cm | −0.337 | −0.640 | −0.200 | −0.284 | 0.583 | | | |
| | 20–30 cm | −0.996 ** | −0.556 | −0.959 * | −0.993 ** | 0.763 | 0.380 | | |
| | 30–50 cm | −0.592 | 0.426 | −0.736 | −0.599 | 0.648 | −0.240 | 0.516 | |
| | 50–70 cm | −0.727 | 0.066 | −0.787 | −0.699 | 0.965 * | 0.374 | 0.677 | 0.809 |

* Indicates that the correlation is significant at $p < 0.05$. ** Indicates that the correlation is significant at $p < 0.01$.

### 3.3. Response of Dynamic Soil Moisture Content to Rainfall

#### 3.3.1. Precipitation Patterns

From July to the end of October 2021, 73 precipitation events were recorded, with a cumulative rainfall of 376 mm during the study period. According to the level of rainfall divided by the Chinese meteorological department, the rainfall of light rain is less than 10 mm in 24 h, the rainfall of moderate rain is 10–25 mm in 24 h, the rainfall of heavy rain is 25–50 m in 24 h, and the rainfall of a rainstorm is more than 50 mm in 24 h. The 64 rainfall events during the monitoring period were divided (Table 5). The most common precipitation event was light rain event (<10 mm precipitation), accounting for 84.93% of all precipitation events. Other rainfall events correspond to moderate rain events and heavy rain events, accounting for 9.59% and 5.48% of the total rainfall events, and no rainstorm events occurred during the monitoring period. The larger rainfall events occurred mainly in July and August, when most of the precipitation was contributed (Figure 3).

**Table 5.** Characteristics of rainfall events.

| Type of Rainfall Event | Rain Frequency | Proportion/% | Rainfall/mm | Contribution Rate/% |
|---|---|---|---|---|
| Light rain | 64 | 87.67 | 109.8 | 30.15 |
| Moderate rain | 5 | 6.85 | 96.4 | 26.47 |
| Heavy rain | 4 | 5.48 | 158 | 43.38 |
| Rainstorm | 0 | 0 | 0 | 0 |
| Total | 73 | 100 | 364.2 | 100 |

#### 3.3.2. Response of Soil Moisture to Rainfall

As can be seen from Figures 4–6, the initial soil water content of each plant community structure type before different rainfall events showed that 0–10 cm was SH > AH > H > ASH; 10–20 cm showed H > AH > SH > ASH; 20–30 cm was AH > SH > ASH > H; 30–50 cm was SH > AH > ASH > H; 50–70 cm showed SH > AH > ASH > H, which was consistent with the spatial and temporal distribution trend of soil water content (Figures 3 and 4). The light rain event occurred from 10:00 to 15:30 on 31 July 2021, lasting 5.5 h, with a total rainfall of 4.6 mm and an average rainfall intensity of 0.84 mm/h (Figure 5). During the light rain event, the soil moisture content of each plant community structure type did not change significantly and failed to recharge soil water effectively, and the soil moisture content of each soil layer remained relatively stable.

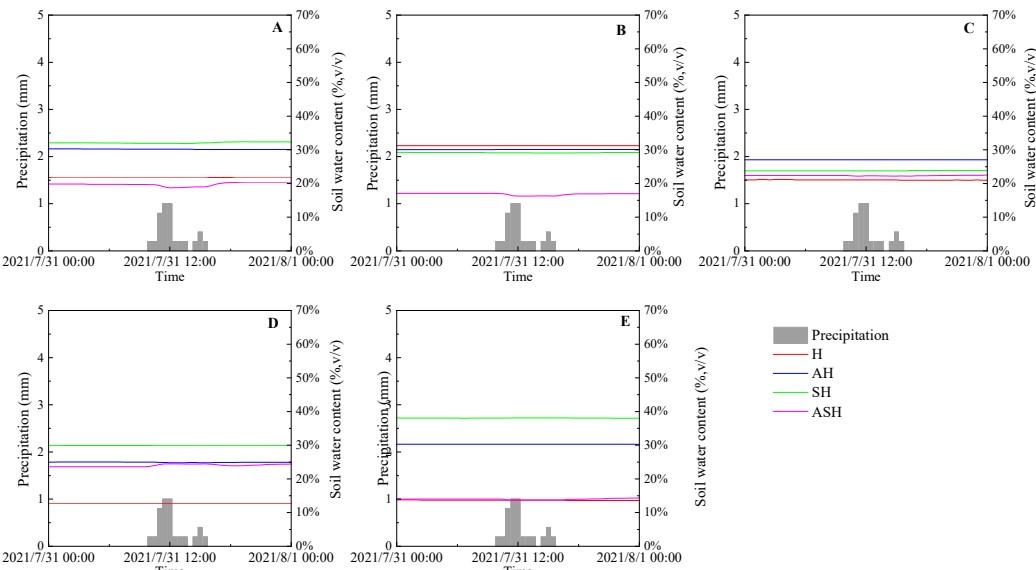

**Figure 5.** The variations of soil moisture content at light rain event. (**A**) 0–10 cm, (**B**) 10–20 cm, (**C**) 20–30 cm, (**D**) 30–50 cm, (**E**) 50–70 cm.

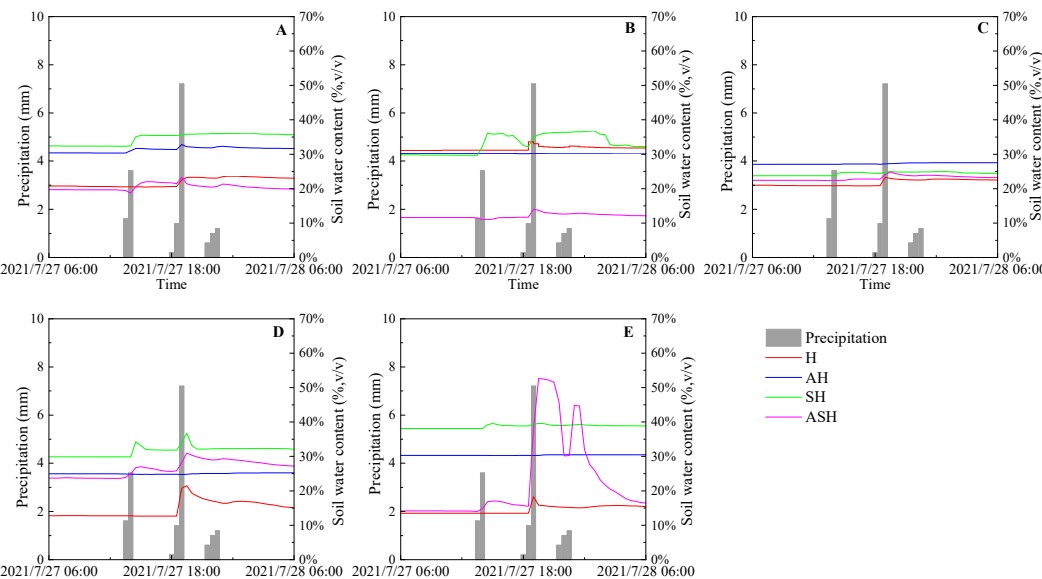

**Figure 6.** The variations of soil moisture content at moderate rain event. (**A**) 0–10 cm, (**B**) 10–20 cm, (**C**) 20–30 cm, (**D**) 30–50 cm, (**E**) 50–70 cm.

The moderate rain event occurred from 13:30 to 22:30 on 27 July 2021, lasting 9 h, with a total rainfall of 16.8 mm. It can be seen that the plant community structure types of H, SH and ASH all responded to the moderate rain event in Figure 6. The AH only 0–10 cm soil layer was responsive to rainfall, and soil moisture content in 10–20, 20–30, 30–50 and 50–70 cm soil layers was relatively stable. The first rainfall lasted for 1 h, and the rainfall intensity was 5.2 mm/h. The response of H and AH to rainfall was not obvious. The increase in soil moisture content of SH and ASH had a lag, with a lag time of 0.5–1 h and a rise time of 0.5–2 h. The second rainfall lasted for 1.5 h, and the total rainfall was 8.8 mm. The soil moisture content of all plant community structure types reached the peak after the second rainfall. The increment of 0–10 cm soil moisture content was SH (3.74%) > ASH (3.66%) > H (2.96%) > AH (2.47%). The increment of soil moisture content in the 10–20 cm soil layer was SH (7.02%) > H (2.44%) > ASH (2.38%) > AH (0.09%). The increment of soil moisture content in the 20–30 cm soil layer was ASH (2.44%) > H

(2.41%) > SH (1.17%) > AH (0.47%). The increment of soil moisture content in the 30–50 cm soil layer was H (8.01%) > ASH (7.41%) > SH (6.91%) > AH (0.15%). The increment of soil moisture content in the 50–70 cm soil layer was SH (38.51%) > H (4.77%) > SH (1.44%) > AH (0.20%). Between the end of the second rainfall and the beginning of the third rainfall, the soil moisture content of each plant community structure type gradually decreased. Then, there was a rebound to varying degrees during the third rainfall. The soil moisture content of the ASH50–70 cm layer increased rapidly during the second and third rainfall. In the event of moderate rainfall, the SH soil moisture content in the 0–10 and 10–20 cm layers increased most obviously, and the ASH soil moisture content in the 20–30, 30–50 and 50–70 cm layers increased most obviously. The AH soil moisture content in each layer of moderate rainfall event did not increase significantly.

The heavy rainfall occurred from 20:30 on 21 July 2021 to 04:30 on 22 July 2021, lasting for 8 h, with a total rainfall of 49.4 mm and an average rainfall intensity of 6.18 mm/h. It can be seen that all community structure types responded significantly to the heavy rainfall event, and each soil layer reaches its peak within 1–3 h after the rainfall begins, in which the soil moisture content of each soil layer in the H and ASH changes greatly (Figure 7). In the event of heavy rainfall, the soil moisture rise period for each plant community structure type was 1–3 h for 0–10 cm soil, and the increase in soil moisture content showed ASH (18.82%) > AH (8.51%) > SH (3.55%)> H (3.09%). The rising period of soil moisture in the 10–20 cm soil layer was 1–2.5 h, and the increase in soil moisture content was ASH (38.00%) > SH (6.97%) > H (2.60%) > AH (2.22%). The increase period of soil moisture in the 20–30 cm soil layer was 1–2 h, and the increase in soil moisture content was ASH (37.25%) > H (21.30%) > AH (4.76%) > SH (1.45%). The increasing period of soil moisture in the 30–50 cm layer was 1–2.5 h, and the increase in soil moisture content was H (38.63%) > ASH (28.87%) > AH (14.48%) > SH (9.22%). The increase period of soil moisture in the 50–70 cm soil layer was 1–2 h, and the increase in soil moisture content was H (41.76%) > ASH (38.95%) > AH (9.31%) > SH (1.92%). In the event of heavy rainfall, the ASH soil moisture content in the 0–10, 10–20 and 20–30 cm layers increased most obviously, and the H soil moisture content in the 30–50 and 50–70 cm layers increased most obviously.

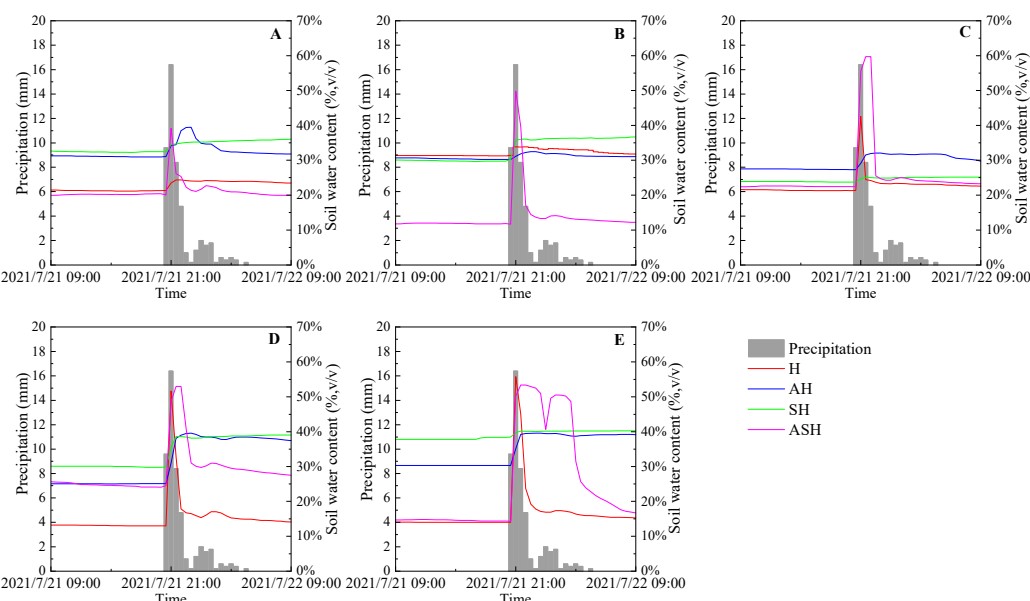

**Figure 7.** The variations of soil moisture content at heavy rain event. (**A**) 0–10 cm, (**B**) 10–20 cm, (**C**) 20–30 cm, (**D**) 30–50 cm, (**E**) 50–70 cm.

## 4. Discussion

### 4.1. Effects of Plant Community Structure Types on Soil Moisture

In this study, due to the limitation of the research scale, there were fewer species in the community structure types. There were 31 species belonging to 18 families and 31 genera (Table 3), among which Compositae, Poaceae, Rosaceae, Urticaceae and Rubiaceae were more concentrated, and the other families were single families and single genera. Arbors and shrubs were confined between a few single species, and the woody layer plants were relatively monotonous, while the herbaceous layer plants occupy a relatively high proportion and are mostly perennial herbs, which was consistent with the survey results of Zhang H et al. [32]. Among the four plant community structure types, H has the largest vegetation coverage, which may be because the grassland ecosystem has relatively sufficient light and water compared with other plant community structure types; thus, the vegetation coverage of H is larger. Species diversity is fundamental to the ecosystem function maintenance and structural stability, which can not only directly reflect the structural characteristics of the community but can also have implications for environmental conditions [16]. Species diversity characteristics of different plant community structure types showed that there were differences in species diversity among different plant community structure types. There were significant differences between AH and H in Shannon–Wiener diversity index, Simpson diversity index and Pielou evenness index. There were significant differences between AH and ASH in all diversity indexes, and significant differences in Pielou evenness index and Simpson diversity index between SH and AH. The Pielou evenness index was significantly higher for SH and ASH than for H. The Margalef richness index, Shannon–Wiener diversity index and Simpson diversity index of different plant community structure types showed H > ASH > SH > AH. The species composition of the arbor layer and herb layer of AH community structure type was single. The arbor layer was mainly dominated by *Juglans regia*, and the herb layer was dominated by *Lolium perenne*, *Stellaria vestita* and *Artemisia argyi*.

Soil moisture as a limiting factor for plant growth in the karst desertification area is closely related to the structural composition and diversity of plant communities; thus, the study of the relationship between species diversity and soil moisture is of great significance for further analysis of the effects of plant community and structure type on soil water content. Through the correlation analysis between species diversity and soil moisture of plant community structure types, we found that the Shannon–Wiener diversity index was significantly positively correlated with Simpson diversity index and Margalef richness index. The Margalef richness index was highly significantly positively correlated with the Simpson diversity index (Table 4). The 20–30 cm soil layer showed a highly significant negative correlation with Margalef richness index and Simpson diversity index, and a significant negative correlation with the Shannon–Wiener diversity index, indicating that soil moisture in the 20–30 cm soil layer of each plant community structure type has a significant effect on plant diversity. The higher the plant diversity index, the lower the soil water content in the 20–30 cm soil layer. This may be related to the type of vegetation and the distribution of the root system, and the heterogeneity of vegetation type and root distribution will lead to the heterogeneity of plant water consumption [33]. It has been found that the increase in plant diversity can intensify the competition among roots and the phenomenon of ecological niche differentiation, so that the roots can fully occupy the soil space, while the composition of vegetation types determines the distribution of roots, and the densely distributed areas consume more soil water [20]. Yang et al. found that soil moisture at 40–60 cm was the main source of water absorption by plant roots when they studied the soil water migration process in the northwest Loess Plateau, and that karst areas differ from the Loess Plateau due to factors such as shallow soil layers and complex habitats [34]. In this study, the increase in plant diversity caused the community roots to form a dense zone in the 20–30 cm layer and to consume more soil water, which was consistent with the conclusion that soil water contributed more to plant water utilization at the depth of 20–40 cm in the karst [35]. There was no significant

correlation between moisture in each soil layer and Pielou evenness index, indicating that plant growth was limited by a number of factors. It has been found that plants can form their own characteristics to adapt the habitat conditions through long-term evolution, in which species are more evenly distributed in the community and in which soil moisture changes have less impact on them [36]. In this study, the vegetation has been growing in the karst fragile ecosystem for a long time, and with the succession process, the community structure types adapted to the habitat conditions were formed. Therefore, there was no significant relationship between the species evenness index and soil moisture.

### 4.2. Temporal–Spatial Variability for Soil Moisture for the Four Plant Community Structure Types

The variation of soil moisture is affected by vegetation transpiration, soil evaporation, topography, and vegetation types [37]. In this study, we found that the soil moisture variation trend of four plant community structure types was consistent with the temporal variation of rainfall (Figure 4). The rainfall events were followed by variation of soil moisture, particularly the heavy rain events (Figure 7). This result is consistent with the results of soil moisture variation with rainfall in different regions, and the stronger rainfall events are often accompanied by larger soil moisture variation [38,39]. Our observation period was within the rainy season of the study area, with moderate rain and heavy rain events mainly concentrated in July and August, accounting for 77.65% of the total rainfall in the monitoring period (Figure 4). The precipitation in the monitoring period was characterized by light rain events (<10 mm), accounting for 87.67% of all rainfall events, and no rainstorm events occurred (>50 mm) (Table 5). The significant difference in soil moisture among different plant community structure types may be due to the difference in evapotranspiration caused by different root distribution of vegetation types and soil physical properties (Figure 3). The large aboveground biomass, strong fertility, deep root distribution and high density of ASH make the water demand for vegetation growth greater than of other plant community structure types, and the strong transpiration consumption results in low soil water content in all layers of ASH. Plant communities can limit soil water evaporation through a microclimate of high humidity and low temperature created under the leaves. The soil water content of AH and SH in the 0–10 cm soil layer is significantly higher than H due to the low height and the lack of dense canopy leaves [40]. In karst areas, tree and irrigation forests can increase the soil porosity through root penetration, so that the water content in the deep soil is higher than that in grassland, and in this study, the soil water content in the 30–70 cm soil depth H showed lower levels, and AH and SH were significantly higher than H [41]. The average soil water content of H was the highest in the 10–20 cm layer, which may be mainly due to the shallow and dense root system of grass, and the release of mucilage and water from the root surface made the water content of the rhizosphere soil higher than that of the non-rhizosphere soil [42]. The soil moisture of the four plant community structure types varied with the rainfall trend in different degrees on a temporal scale, which was consistent with the rainfall trend, but with differences at different soil layers. Therefore, we consider rainfall and plant community structure type to be the dominant factors influencing the temporal variation and spatial distribution of soil moisture.

### 4.3. Differences in Responding Soil Moisture of Four Plant Community Structure Types to Rainfall

The response process of soil moisture to rainfall is complex, and its dynamics are affected by community characteristics, soil properties, topography and other factors. Our results showed that the temporal variation trends of soil moisture for plant community structure type were consistent with that of rainfall events, and the dynamic variation of soil moisture content of plant community structure types was significantly affected by rainfall during moderate rainfall events and heavy rainfall events, indicating that rainfall characteristics were an important factor affecting soil moisture recharge. Before the occurrence of different rainfall events, the initial soil water content of different plant community structure types was consistent with the temporal and spatial distribution

characteristics. The differences in evapotranspiration caused by different root distributions of vegetation types and soil physical properties resulted in heterogeneity of soil moisture in each soil layer of different plant community structure types. Among the rainfall events of different magnitudes, the light rainfall events had the least influence on the change in soil moisture content. The soil moisture content of each soil layer of H, AH, SH and ASH was in a relatively stable state, which may be due to the rapid water loss caused by vegetation interception, surface vegetation cover and plant transpiration, and it failed to effectively replenish soil moisture. With the increase in rainfall level, the dynamic changes in soil moisture content of the four plant community structure types were significantly responsive to rainfall. In the event of moderate rainfall, ASH has the greatest amount of soil moisture recharge in the middle and deep layers soil, which might be related to the lower soil bulk density and larger porosity of ASH plant community structure type (Table 1). In addition, the underground intricate root network of arbors, shrubs and herbs interpenetrate the soil of ASH community structure, providing nesting for soil animal activities, increasing soil pores and providing preferential flow paths, so that moisture can enable adequate infiltration [43]. In the event of heavy rainfall, the soil moisture content of the four plant community structure types changed obviously, and ASH soil moisture content increased more in all soil layers. The slower increase rate of soil moisture in each soil layer of SH may be related to the vegetation coverage and structural composition. The vegetation coverage of the SH canopy and surface is low, and the canopy and branches have less interception of rainfall, resulting in raindrops hitting the ground directly, damaging the surface structure and reducing soil infiltration during heavy rain events.

There are limitations in the scale of the study and the setting of observation points in this paper, and future research should expand the scope of plant community structure investigation and add more monitoring points to study the relationship with soil moisture. In addition, only the correlation between species diversity of plant community structure and soil moisture content and the response of soil moisture content to rainfall were considered, and the effects of plant community structure types on soil nutrients were ignored in this study. It has been shown that species diversity is correlated with the carbon and nitrogen cycle, and different plant community structures are closely related to the loss of soil nutrients [44,45]. In the future, research on the effects of synergistic changes of rainfall and nutrients on the structural characteristics of plant communities should be strengthened in order to gain a deeper understanding of vegetation development patterns in ecologically fragile areas and to provide a reference for ecological restoration and revegetation.

## 5. Conclusions

The plant community mainly belongs to Compositae, Gramineae, and Rosaceae and is dominated by perennial herbaceous. The Margalef richness index, Shannon–Wiener diversity index and Simpson diversity index of different plant community structure types showed H > ASH > SH > AH. Soil moisture content in the 20–30 cm soil layer was highly significantly negatively correlated with the Margalef richness index and the Simpson diversity index and was significantly negatively correlated with the Shannon–Wiener diversity index. Soil moisture variation in the four plant community structure types was mainly controlled by precipitation and above-ground community structure types. The temporal variation of soil moisture was consistent with the trend of rainfall events, and the deep soil moisture had a significant lag to precipitation events. Soil moisture content of H, AH, SH and ASH was significantly different. The maximum average soil moisture content of AH, SH and ASH was in middle and deep soil, while H was in shallow soil. Light rain events had little effect on soil moisture of plant community structure types and failed to effectively replenish soil moisture. With the increase in rainfall level, the dynamic change of soil moisture content was significantly in response to rainfall.

**Author Contributions:** Conceptualization, X.G.; methodology, X.G. and C.W.; software, X.G. and D.L.; validation, X.G.; formal analysis, X.G.; investigation, X.G. and D.L.; resources, X.G.; data curation, X.G.; writing—original draft preparation, X.G.; writing—review and editing, X.G. and D.L.; visualization, X.G.; supervision, K.X.; project administration, K.X.; funding acquisition, K.X. All authors have read and agreed to the published version of the manuscript.

**Funding:** This research was funded by the Key Science and Technology Program of Guizhou Provence: Poverty Alleviation Model and Technology Demonstration for Ecoindustries Derived from the Karst Desertification Control (no. 5411 2017 QianKehe Pingtai Rencai), the China Overseas Expertise Introduction Program for Discipline Innovation: Overseas Expertise Introduction Center for South China Karst Eco-environment Discipline Innovation (D17016), and the Project of National Major Research and Development Program of China in the 13th Five-year Plan: Ecological Industry Model and Integrated Technology Demonstration of the Karst Plateau-Gorge Rocky Desertification Control (2016YFC0502607).

**Data Availability Statement:** Not applicable.

**Acknowledgments:** The authors would like to thank each editor for their contributions with this paper, which enrich it, as well as for the help provided by the local residents in the land survey, the experimental equipment and good environment provided by the State Engineering Technology Institute for Karst Desertification Control, School of Karst Science, Guizhou Normal University.

**Conflicts of Interest:** The authors declare no conflict of interest.

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
