# Peer review of "Study on Soil Moisture Characteristics in Southern China Karst Plant Community Structure Types"

_forests, doi:10.3390/f14020384_

Round 1

Reviewer 1 Report

It is very significant to study on species diversity characteristics of different plant community structure types and their correlation relationships. The results provide a reference for the effective utilization of soil and water resources and the restoration of vegetation in Southern China Karst region. This study is intriguing and has scientific potential. However, it has many flaws and weaknesses that need to be addressed in more detail.

Point 1: Line 89-93, the paper should clearly provide the objects of this study and key problems want to be solved. But the manuscript did not clearly express the purpose of the study in Introduction.

Point 2: Line 126, how many times did the author sampling from July to August 2021? please tell us clearly. Because, we don’t know if the plant diversity maybe changed or not?

Point 3: Lin 241-243, “The 20-30 cm soil layer showed highly significant negative correlations (P < 0.01) with the Margalef richness index and Simpson diversity index, and showed significant negative correlation with the Shannon-Wiener diversity index (P < 0.05). the result is very interesting. This paper showed that plant community structure types, and diversity characteristics can directly affect soil water content. Why the negative correlations?

Point 4: Line 260, Response of soil moisture to rainfall, The light rain, the moderate rain and the heavy rainfall events all effect soil water content. But why soil moisture content different in different plant community structure types? please explain.

Point 5: Line 358-363: The higher the plant diversity index, the lower the soil water content in 20-30cm soil layer. This may be related to the type of vegetation and the distribution of the root system. How the type of vegetation and the distribution of the root system effect soil water content? the author should explain.

Author Response

Dear reviewer,

I have re-thought and revised the article in depth according to your questions and suggestions, and I have given detailed replies to each question. Thank you for your suggestions on the article. I have learned a lot. Please refer to the attachment for specific answers.

Reviewer 2 Report

1.       Too many literature research results of Studies on the relationship between plant community structure type and soil moisture out in non-karst areas were elaborated. The scientific questions are not clear enough, and it is suggested to elaborate according to the research object and content in combination with relevant literature

2.       The name of the tree shall be given in Latin name at the same time, such as Masson pine , rhododendron forest etc. Masson pine and Horsetail pine Should be unified.

3.       Please confirm whether the China map in Figure 1 conforms to the use specifications.

4.       Table 2 shows the main plant species composition of plant community. What are the plant species of different structure types? Latin names of plants should be italicized

5.       In “3.2.1. Vertical distribution characteristics of soil moisture” ,the variation rule of soil moisture with soil depth in different vegetation types was analyzed, and it was suggested that the influence of different vegetation types on the vertical distribution of soil moisture should be clarified in combination with further characteristics of different vegetation types.

6.       Line 11: delete “ e the ” .

7.       Line 13: “research object” should be  “research objects”.

Author Response

(The authors gave the same response as above.)
